# The Stimulation of Macrophages by Systematical Administration of GM-CSF Can Accelerate Adult Wound Healing Process

**DOI:** 10.3390/ijms231911287

**Published:** 2022-09-25

**Authors:** Jing Zhang, Liyuan Jia, Hanxue Zheng, Juantao Feng, Sili Wei, Juan Li, Jihong Cui, Fulin Chen

**Affiliations:** 1Laboratory of Tissue Engineering, College of Life Sciences, Northwest University, Xi’an 710069, China; 2Provincial Key Laboratory of Biotechnology of Shaanxi, Northwest University, Xi’an 710069, China; 3Key Laboratory of Resource Biology and Biotechnology in Western China, Ministry of Education, School of Medicine, Northwest University, Xi’an 710069, China

**Keywords:** macrophage, skin defect, wound repair, L-clod, GM-CSF

## Abstract

Skin wound repair remains a major challenge in clinical care, and various strategies have been employed to improve the repair process. Recently, it has been reported that macrophages are important for the regeneration of various tissues and organs. However, their influence on wound repair is unclear. Here, we aimed to explore whether macrophages would participate in the wound healing process and to explore new possibilities of treatment for skin defects. We firstly created a mouse full-thickness skin defect model to observe the distribution of macrophages in the regenerating tissue and then detected the influence of macrophages on skin defect repair in both macrophage-depletion and macrophage-mobilization models. We found that the number of macrophages increased significantly after skin defect and persisted during the process of wound repair. The regeneration process was significantly prolonged in macrophage-depleted animals. RT-qPCR and ELISA assays further demonstrated that the expression of growth factors was perturbed in the regenerating tissue. The activation of macrophages by granulocyte-macrophage colony-stimulating factor (GM-CSF) injection could significantly improve wound healing, accompanied with an upregulation of the expression of various growth factors. In conclusion, the current study demonstrated that macrophages are critical for skin regeneration and that GM-CSF exhibited therapeutic potential for wound healing.

## 1. Introduction

Large full-thickness skin defects [1,2] remain a significant challenge in clinical care. At present, autologous or allogeneic skin grafting is still the main choice for clinicians. Tissue-engineered skin is an optimized alternative for the treatment of skin defect [3,4]. However, they all have their shortcomings due to factors such as limited sources, high storage cost, complex surgery, etc. [5,6,7,8,9].

Local applications of recombinant growth factors have long been considered as an efficient method for tissue repair and regeneration [10,11]. Recombinant human bone morphogenetic proteins (BMPs) have been approved by the FDA for orthopaedic application. However, the application of BMPs during pregnancy is prohibited because of the unclear long-term genetic effects, and cautious application is also advised in children [12]. Growth factors, especially fibroblast growth factors (FGFs), may stimulate cell proliferation, differentiation, and migration, which improve the wound healing process. However, growing evidence has implicated FGFs in the genesis of cancers after repeated local application due to their strong mitotic activity [13].

In recent years, there have been reports that the presence of macrophages is crucial for tissue regeneration. The depletion of macrophages before salamander limb amputation affected collagen deposition, causing excessive fibroplasia and a complete blockage of blastemal formation [14]. The engagement of regenerative processes may be partly related to the early arrival of macrophages to the regenerating local site. A delayed redevelopment associated with an apparent reduction in surface vasculature was also observed when the depletion of macrophages was performed after blastemal formation. In addition, macrophages presenting in stratum were associated with the development of chondrification centers and provided pro-anabolic support to soft callus formation. In the late anabolic phase, the presence of predominantly resident macrophages, especially osteomacs, showed important contributions to endochondral callus formation [15]. Furthermore, macrophages were able to migrate toward the lesions of blood vessels to help efficiently improve the extension of the two broken endothelial ends by physical contact and mechanical forces [16]. In addition, macrophages were found to be sensitive to hypoxic environments and could quickly respond to a peripheral nerve injury. By secreting vascular endothelial growth factor A (VEGF-A) within the ‘bridge’ of the injured nerve tissues, they stimulated the formation of blood vessels which acted as a scaffold for the migration of Schwann cells. Because Schwann cells play a guiding role in axon regrowth, macrophages were thus considered to have an important indirect role in the process of nerve regeneration [17].

The effect of macrophages and their mechanism during the wound healing process is unknown. We hypothesize that macrophages play a critical role in skin wound repair and that stimulating the proliferation and differentiation of macrophages could effectively improve the wound healing process. Herein, we investigated the distribution of macrophages throughout the wound healing process. To delineate the functional contributions of macrophages to skin damage repair, we used a reversible model of macrophage depletion. We also used appropriately timed delivery of granulocyte-macrophage colony stimulating factor (GM-CSF) to enhance the proliferation and differentiation of macrophages during the whole regeneration process. We aimed to mainly find the role of macrophages during the wound healing process and explore new possibilities of treatment for skin defect.

## 2. Results

### 2.1. Macrophages Presented throughout the Whole Process of a Full-Thickness Incision Skin Defect

We firstly performed a full-thickness defect operation on the back of BALB/c mice, and we then checked the distribution of macrophages by IF staining using an F4/80 antibody (Figure 1b). The results showed that macrophages were evenly distributed in the dermis layer of the intact mouse back skin (0D). Upon injury, the number of macrophages in the damaged skin tissue was significantly increased and reached a peak at 5D (Figure 1d, upper), and a large portion of them was present in the granulation tissue. With the completion of the repair process, macrophages shifted to the bottom region of the epidermis and the superficial part of the dermis, continuing to maintain a relatively high quantity.

### 2.2. The Population of Macrophages Was Decreased after the Injection of Clodronate Liposomes

In the present study, we introduced liposome-carried dichloromethylene diphosphonate, specifically called clodronate liposome (L-clod), by tail vein injection to eliminate macrophages in mice (Figure 1a).

To determine the cell type-specific effect, the dorsal skin of either L-PBS-injected (PBS liposome) or L-clod-injected mice was incised, and samples were collected at 5D, 10D, 15D, and 20D. Compared with the control group, the administration of L-clod efficiently prevented the accumulation of macrophages at the wound sites (Figure 1c). The number of F4/80^+^ cells in mice of the experimental group was significantly lower than that in the control mice, and the depletion effect lasted until the end of the repair process (Figure 1d, lower). The difference in the number of macrophages between the L-clod and the control L-PBS groups at each time point was similar to the statistical results of L-clod compared with the untreated Normal groups. These results indicated that continuous tail vein injection of L-clod in mice led to a low expression of both resident and migration macrophages at the local injury site.

### 2.3. The Depletion of Macrophages Resulted in Delayed Wound Closure and Prolonged Re-Epithelialization

The influence of the macrophage depletion treatment on wound healing was assessed. The healing area was significantly smaller at both 5D and 10D in the L-clod-treated mice than in the control mice (Figure 2, Table 1). By the end of 15D, mice from the control group had completed the wound closure process, while L-clod-treated mice had achieved the whole healing process at 20D.

H&E staining was then used for detailed analysis. The control mice showed a higher wound closure percentage, with all wounds closed at 10D, whereas mice from the treatment group displayed significantly delayed wound closure at 5D (19.73 ± 6.05% L-clod vs. 38.50 ± 1.95 L-PBS, *p* < 0.01, Figure 3), and all of them failed to complete wound closure by the end of 10D (67.63 ± 17.18% L-clod vs. 100% L-PBS, *p* < 0.001, Figure 3a–g). The repair trends of the two groups obtained by H&E staining corroborated the macroscopical observations that the repair rate of the experimental group was significantly decreased.

Collagen deposition in lesion tissues from both of the two groups was analyzed using Masson’s trichrome staining. Collagen started to accumulate as early as 5D in the skin tissues in both groups (Figure 3h). During the healing process, the collagen volume fraction (CVF) of tissues in the L-clod treatment group was significantly lower than in the control group (Figure 3g, right). However, as the wound was completely healed, the CVF in both groups returned to a same level. These results showed that the injection affected the recovery of the wound but not the final CVF.

### 2.4. The Expression Levels of Regeneration-Associated Genes and Proteins after Macrophage Depletion

To study the mechanism of how macrophage depletion affected the wound healing process, we evaluated the expression of the genes/proteins known to be involved in the skin repair process. By examining molecular profiles at 3D, we detected decreased expression of the macrophage chemoattractant *Il8*, the pro-inflammatory genes *Tnfα* and *Csf1r,* and the pleiotropic cytokine *Il6*, as well as increased expression levels of the antimicrobial peptide *Camp* in tissue homogenates in the treatment group (Figure 4a). During the healing process, the regeneration-associated gene profiles were modulated after the injection of L-clod (Figure 4b). The key regulator of angiogenesis, *Vegfa*, which is predominantly expressed by macrophages [18,19], was observed to be downregulated in mice injected with L-clod. The depletion of macrophages also caused diminished expressions of *Tgfb1*, *Mmp2*, and *Mmp3* during the healing process. Relatively low expressions of *Snail*, *Fgf2*, and *Mmp9* at 3D were also detected. The results of ELISA experiments showed that the protein expression profiles of these growth factors were consistent with the expression profiles at the gene level (Figure 4c).

### 2.5. The Injection of GM-CSF Resulted in an Upregulation of the Quantity of Macrophages in Both Peripheral Blood and Skin Tissue

Based on the above results indicating that macrophages did participate in the process of skin repair, we assumed that artificially enhanced macrophage proliferation and differentiation would probably accelerate the skin healing process. GM-CSF is well known as a pro-inflammatory factor that promotes the recruitment of myeloid cells for their survival and activation at inflammatory sites [20]. We thus firstly subcutaneously injected GM-CSF into mice and tested its stimulating effect on macrophages in the peripheral blood by flow cytometry. FACS analysis showed that the percentage of macrophages was significantly increased after only one dose of GM-CSF injection, and another dose of injection contributed to the enhancement of the drug effect (Figure 5a,b, left). We then analyzed the mobilization effect of GM-CSF by quantifying macrophages in skin tissues from both the GM-CSF injection group and the control group at different time points, and we found that the skin tissues of GM-CSF injected mice contained more F4/80^+^ cells than those of the control group at each time point (Figure 5b, right and Figure 5c). These results indicated that the GM-CSF injection treatment resulted in a rising number of macrophages in both the peripheral blood and the injured skin tissue at the wound site.

### 2.6. The Treatment of GM-CSF Injection Accelerated Wound Healing Process by Upregulating Regeneration-Associated Genes

To further investigate the influence of the GM-CSF injection treatment on the skin injury repair process, we compared the healing rate between the control and treatment groups. The healing process of the GM-CSF injected mice was significantly accelerated compared to that of the control mice. At 6D and 9D, the recovery area of the experimental mice was significantly larger than that of the control mice (Figure 6, Table 2). Wounds of experimental mice healed at 12D, compared to 15D in the control mice.

Furthermore, we examined the rate of re-epithelialization by H&E staining. The results showed that the GM-CSF group completed re-epithelialization at 9D, while the control group acquired intact dorsal neo-skin by the end of 12D (Figure 7a–i).

Interestingly, Masson’s trichrome staining showed that after the injection of GM-CSF, the collagen content in the skin tissues of mice from the experimental group was much lower than that of the control group during the whole regeneration process (Figure 7i, right). In addition, after accomplishing wound healing, the CVF in skin tissues from GM-CSF-injected mice was still less than that of the control mice (*p* < 0.001), and the collagen structure was much looser (Figure 7j), resembling the original collagen condition of the undamaged skin.

The treatment of GM-CSF injection also contributed to an upregulation of the expression of *Snail*, *Vegfa,* and *Mmps*, but the *Tgfb1* signaling pathway seemed to be unaffected (Figure 7k). It is worth noting that the mice from the treatment group exhibited a prolonged and over-expressed level of *Vegfa* from 9D to 15D.

## 3. Discussion

Skin is extremely sensitive and vulnerable to external damages, though it possesses a strong self-repairing ability. Large skin trauma usually requires immediate intervention and treatment by artificial means. However, the existing treatment methods have various deficiencies. The process of skin regeneration is in general divided into four spatially and temporally overlapping steps: homeostasis, inflammation, proliferation, and remodeling [21,22]. At the very first phase of wound healing, phagocytic cells and other lymphocytes infiltrate into the wound area to clear debris and bacteria from the wound [21]. However, whether these cells are related to tissue regeneration is still under debate. On the other hand, it has been reported that in addition to their phagocytic ability, macrophages can participate in the regeneration of multiple organs. However, their specific effects and mechanisms regarding skin damage repair have not yet been reported in detail.

In this study, we observed that macrophages distributed in the newly regenerated granulation tissue, and their quantity changed throughout the healing process. The number of macrophages in the wound area increased as an early response to skin injury and reached to a peak at 5D. To test a hypothesis that the presence of macrophages had an impact on skin damage repair, we then created a macrophage-depletion system by injecting L-clod. After injection, the quantity of macrophages was significantly decreased, while other white blood cells (WBC) maintained their original expression levels (data not shown). These data are consistent with others’ work indicating that the administration of L-clod efficiently knocks down the number of macrophages in the organism without affecting the recruitment of neutrophils and other lymphocytes [23,24]. This treatment caused a detrimental effect on skin repair, such that mice of the L-clod injected group had a delayed recovery rate and re-epithelialization compared with the control mice.

Therefore, we next detected the expression levels of inflammation-related factors in the skin tissues of the two groups at different time points. We observed that the treatment blocked the proliferation and differentiation of macrophages, as a lower expression level of the macrophage surface marker *Csf1r* was detected. The abnormal expression of these pro-inflammatory factors might consequently lead to the failure of sufficient inflammation response activation. As inflammation is a necessary step of the healing process and is important for suppressing exogenous antigens [11,21,25], its inadequate activation might lead to insufficient debris clearance and disrupt the initiation of the following regeneration phase.

In the study of regeneration-related genes’ expression, we found that samples from the experimental group displayed a decreased expression level of both TGFβ1 and MMPs from the early stage and sustained a relatively low level until the end of the regeneration process. The activation of the TGFβ signaling pathway is important for the induction of proliferated fibroblasts to promote the migration of themselves and keratinocytes to wound sites [26,27,28]. The secretion of MMPs helps to break down the extra-cellular matrix (ECM) [29], creating a pro-regenerative microenvironment for epidermal stem cells detaching from niches to differentiate and proliferate. Both TGFβ1 and MMPs are crucial for damaged skin to acquire its final integrity. Lacking TGFβ1 and MMPs would lead to a delayed proliferation and migration of epidermal cells to the local injury site and consequently result in the failure of wound closure in macrophage-depleted mice. Although a concomitant decrease in the expression profile of MMPs was observed on both the transcriptional and translational levels, collagen deposition in the newly formed skin tissues of the two groups was not affected. However, delayed collagen deposition was discovered, suggesting that abnormal expression of MMPs might influence the speed of the process of collagen construction but not the CVF.

An insufficient expression of VEGFA was also detected during the recovery process of the macrophage-depleted mice. VEGFA has been shown not only to be involved in the regulation of vasculogenesis, but also to participate in the differentiation and organization of endothelial progenitor cells into a primary capillary plexus [30]. Besides, it is crucial to the formation of new capillaries from preexisting vessels. Hence, the lack of VEGFA signified a late formation of the new blood vessels, which led to the deficient delivery of essential nutrients and oxygen to mitotically active cells [31,32], and thereby delayed healing progress.

In addition to these growth factors, the results indicated that the expression level of SNAIL was affected. It is well known that SNAIL is a transcription factor that is responsible for the induction of epithelial–mesenchymal transition (EMT) and the activation of the downstream biological process related to various signaling pathways [33,34]. This abnormal expression of SNAIL at the early stage possibly indicated a defective activation of EMT, which might lead to imperfect cell motility and result in prolonged wound closure.

GM-CSF is a cytokine that induces the activation of monocytes/macrophages in vitro [20,35]. We thus introduced GM-CSF to verify whether stimulating the proliferation and differentiation of macrophages could accelerate the wound healing process. The flow cytometry results showed that one dose injection could induce an efficient effect by increasing the number of macrophages in mouse peripheral blood. Accompanied with this augmentation, a larger recovery area was detected in GM-CSF-treated mice. Although the gene expression profile of *Tgfb1* was not affected, the elevated expression of *Snail* as early as 3D signified a quick engagement of EMT, suggesting that the proliferation and differentiation process of regenerative cells had taken place. The injection of GM-CSF also affected the quantity of collagen deposition. The CVF of mice from the GM-CSF treated group was less than that of the control group. In addition, the collagen structure of the experimental mice acquired a morphology similar to undamaged skin, suggesting that the injection of GM-CSF might be beneficial to avoiding massive scar formation. Considering that the expression of *Mmps* was increased, these lyases might contribute to optimizing the precise wound healing process. We also observed a prolonged higher expression level of *Vegfa* which was accompanied by an improved angiogenesis process. The neo-vessels could as a consequence provide passageways for delivering parts of cytokines, chemoattractants, growth factors, and regeneration-associated cells to the injury sites, thus accelerated the healing process.

## 4. Materials and Methods

### 4.1. Animal Maintenance and Full-Thickness Skin Defect Study

Four–six-week-old male BALB/c mice (weight 25–30 g), bought from the laboratory animal unit of the Xi’an Jiaotong University Health Science Center, were housed in a controlled environment (25 ± 3 °C, 40% ± 5% relative humidity, 12 h light-dark cycle) and maintained in accordance with the guidelines provided by the Institutional Ethics Committee of Northwest University.

For wound repair analysis, a rounded (10 mm in diameter) full-thickness incision was created on each mouse’s back after anesthetization by intraperitoneal injection of pentobarbital sodium (100 mg/kg body weight) at 0 days post-injury (D/dpi for short). At each time point, changes in the wound site (n = 5) were firstly recorded by camera, and animals were then sacrificed after overdose anesthesia. Tissue samples were collected and embedded in paraffin for further analysis.

The wound area was measured using the Image J software (Version 1.53e, Wayne Rasband and contributors, National Institutes of Health, Bethesda, MD, USA), and the percentage of recovery area rate was calculated using the following formula, as previously described [36]:Recovery area (%) = [(A0 − Ad)/A0] × 100%
(A0: wound area at 0 dpi; Ad: wound area at sample collecting dpi)

### 4.2. Histology and Immunohistochemistry (IHC)/Immunofluorescent (IF) Staining

Embedded samples were sectioned at a thickness of 8 µm (*n* ≥ 3). The structure of skin wounds was assessed using H&E and Masson’s trichrome (Solarbio Life Sciences, Beijing, China) staining, according to the standard methods [37]. IF staining was performed using fluorescent-labeled secondary antibodies and DAPI (Roche, Basel, Switzerland)/PI (Abcam, Cambridge, United Kingdom). HRP-labeled secondary antibodies were used for IHC staining, and sections were finally counterstained with hematoxylin. IF staining slides were visualized by a laser scanning confocal microscope (Olympus Fluoview™, Tokyo, Japan), while the others were photographed under a digital microscope (Zeiss, Jena, Germany).

### 4.3. Macrophage Depletion

According to the manufacturer’s instruction, clodronate liposomes were intravenously injected (100 µL/mouse) into the caudal vein at −2, 0, 4, 8, 12, 16 dpi, respectively. Control mice received an equal volume of PBS liposomes at the same time points (*n* ≥ 3). Both clodronate and PBS liposomes were purchased from clodronateliposomes.com.

### 4.4. RNA Isolation and Quantitative Gene Expression

Total RNA from mouse dorsal skin tissue was isolated (*n* ≥ 3) using RNAiso Plus (TaKaRa Bio, Kusadzu, Japan) and then reversed using a Transcriptor cDNA kit (Roche, Basel, Switzerland). Quantitative RT-PCR was performed, and mouse α-actin was used as an internal control. Primers used in this study are shown in Table 3.

### 4.5. Protein Isolation and Quantification

Tissue samples (*n* ≥ 3) were rinsed several times and then ground into homogenates in RIPA (ThermoFisher Scientific, Waltham, MA, USA) added with PMSF (Sigma-Aldrich, St. Louis, MO, USA) to extract total proteins. Enzyme-linked immunosorbent assay (ELISA) was performed to detect protein expression levels. The standard sample of tested proteins was used as an internal control.

### 4.6. Macrophage Mobilization

GM-CSF (Tebao Bioengineering, Xiamen, China) was subcutaneously injected into the cervical-dorsal skin of mice (100 ul/mouse) to temporally mobilize macrophages. For wound healing study: a total of 10 doses were injected every other day from −4 dpi until the end of the regeneration process, that is, at −4D, −2D, 0D, 2D, 4D, 6D, 8D, 10D, 12D, and 14D. Control mice received an equal volume of saline solution at the same time points (*n* ≥ 3). For FACS analysis, the GM-CSF-5D group received two doses at 1D and 3D, while the GM-CSF-3D group received a single dose at 3D, and the control group received an equal dose of saline at 1D and 3D (*n* ≥ 3).

### 4.7. FACS Analysis

Blood samples (*n* ≥ 3) were collected at 5D in EDTA tubes and were incubated with either fluorescent-conjugated CD11b or F4/80 antibody (as single positive control) or their isotypes (as a negative control), as well as both fluorescent-labeled CD11b and F4/80 antibodies. Red blood cells were then lysed by adding 1× cell lysis buffer (Tiangen, Beijing, China). Samples were examined on a BD FACSAria III flow cytometer. Results were analyzed using the FlowJo software (Version 10.0.7r2, BD company, Ashland, OR, USA).

### 4.8. Antibodies and Cytokine Detection

The monoclonal antibodies used were: rabbit anti-F4/80 (SP115; Abcam, Cambridge, United Kingdom); rabbit anti-CD11b (EPR1344; Abcam, Cambridge, United Kingdom); mouse anti-VEGFA (VG-1, Abcam, Cambridge, United Kingdom); rabbit anti-alpha tubulin (EP1332Y, Abcam, Cambridge, United Kingdom); FITC rat anti-CD11b (M1/70; BD Pharmingen™, San Diego, CA, USA); FITC rat IgG2b kappa Isotype Control (eB149/10H5; eBioscience™, Santa Clara, CA, USA); Alexa Fluor^®^ 647 rat anti-F4/80 (T45-2342, BD Pharmingen™, San Diego, CA, USA); and Alexa Fluor^®^ 647 rat IgG2a kappa Isotype Control (R35-95, BD Pharmingen™, San Diego, CA, USA).

The polyclonal antibodies used were: rabbit anti-neutrophil elastase (Abcam, Cambridge, United Kingdom); and rabbit anti-beta actin (Abcam, Cambridge, United Kingdom).

Cytokine protein analysis experiments were assayed using mouse ELISA kits (Mlbio, Shanghai, China) according to the manufacturer’s instructions.

### 4.9. Statistical Analysis

SPSS (IBM, Armonk, NY, USA) was used for statistical analysis. Quantitative data were presented as means ± SEM. Student’s *t*-test was used for comparisons between the control and experimental groups. Values of *p* < 0.05 were considered as significant differences.

## 5. Conclusions

Taken together, the injection of GM-CSF resulted in an upregulation of the expressions of various genes associated with cell proliferation, differentiation, and migration, as well as the acceleration of the formation of their transportation systems. As a consequence, mice from experimental groups acquired a faster re-epithelialization speed than that of the control group, and the proportions of their open wound areas were significantly smaller at the same time points (6D, 9D). These results elucidated that the enhancement of the recruitment of monocytes would in a way improve the wound healing process by modulating key genes associated with regenerative pathways. Our current study provided new insights into the development of new strategies and therapies in the future research of skin.

## Figures and Tables

**Figure 1 ijms-23-11287-f001:**
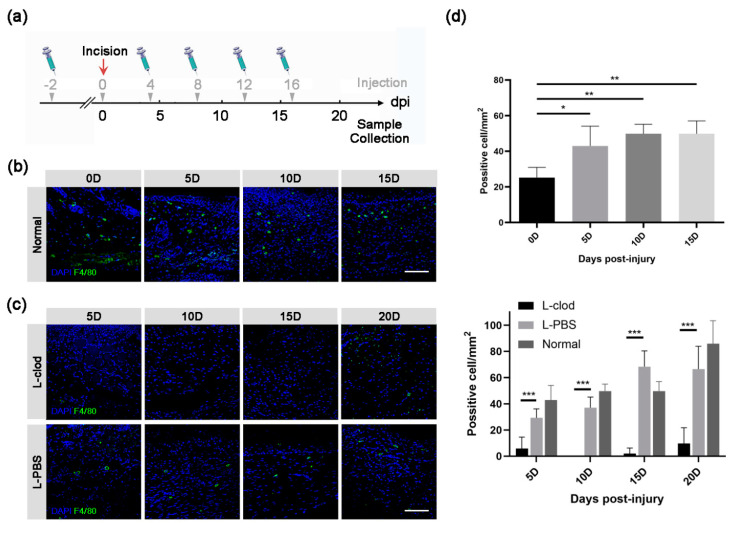
**The distribution of macrophages during wound healing process before or after the depletion treatment.** (**a**) Schematic diagram of macrophage depletion strategy. Gray arrowheads and numbers above the axis indicate L-clod or L-PBS injection time points, while black numbers below represent sample collecting days. The full-thickness skin incision was done at 0 dpi, which is represented as a red arrow. Macrophages in lesion skin tissues of either untreated (**b**) or L-clod- and L-PBS-treated (**c**) mice were detected by anti-F4/80 antibody (scale bar: 80 µm) and counted. The statistical result is presented as mean ± SD in subfigure (**d**) (upper: untreated, lower: L-clod-treated, L-PBS-treated, and untreated Normal groups. *n* ≥ 3. * *p* < 0.05, ** *p* < 0.01, *** *p* < 0.001).

**Figure 2 ijms-23-11287-f002:**
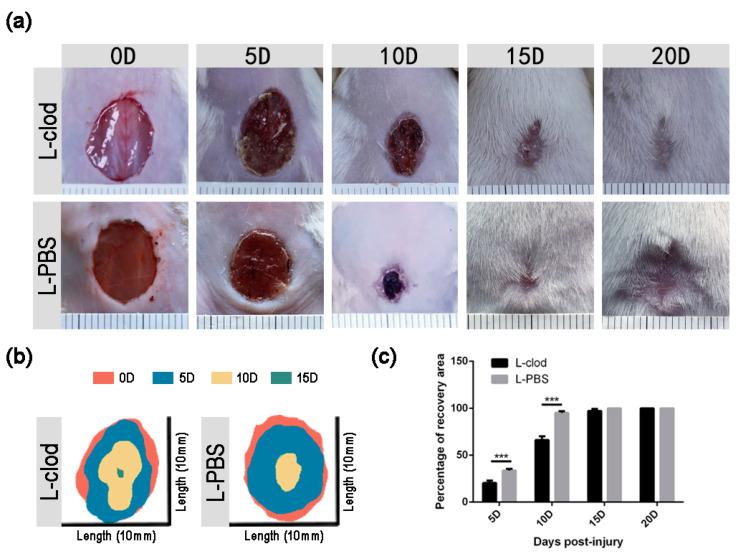
**Wound healing in L-clod and L-PBS-injected mice.** (**a**) Macroscopic appearance of wound sites from mice treated with either L-clod or L-PBS at different time points. (**b**) Schematic diagram of injury site changes over time. (**c**) Quantification of the recovery rate. The open wound area was measured using Image J and represented as percentage of recovery area as described in the Materials and Methods section (*n* ≥ 3. *** *p* < 0.001. Data presented as mean ± SD).

**Figure 3 ijms-23-11287-f003:**
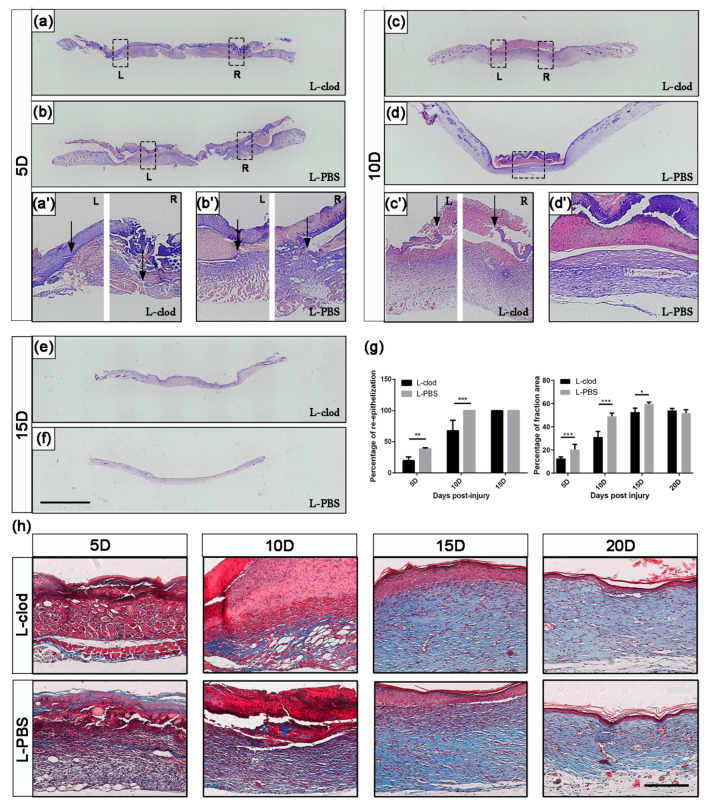
**Analysis of re-epithelialization and collagen deposition after macrophage depletion.** Tissues of wound sites from L-clod- and L-PBS-injected mice were collected at 5D, 10D, and 15D and stained with H&E ((**a**–**f**), scale bar: 2.5 mm). Dashed boxes mark inset showing high-magnification images (**a’**–**d’**). Black arrows indicate ends of migrating epithelial tongues. The statistics of quantification of the percentage of re-epithelialization is presented as mean ± SD in subfigure (**g**), **left** (*n* ≥ 3–5. * *p* < 0.05, ** *p* < 0.01, *** *p* < 0.001). (**h**) Sections of wound area at indicated time points were stained with Masson’s trichrome (scale bar: 200 µm), and the statistics of collagen deposition area are presented as mean ± SD in subfigure (**g**), **right**. Collagen area stained in blue was firstly measured by Image J, and the percentage of fraction area (Collagen Volume Fraction, CVF) was then calculated using the following formula: CVF = (blue area/total skin area) × 100% (*n* ≥ 3. *** *p* < 0.001).

**Figure 4 ijms-23-11287-f004:**
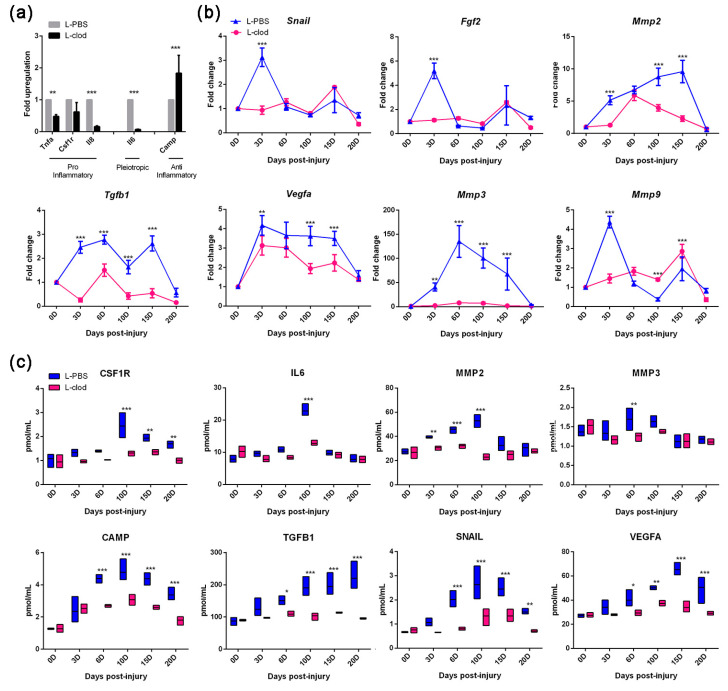
**Gene and protein expression profiles of mice after treatments during wound healing process.** Changes in gene expressions at different time points were assessed by RTq-PCR. (**a**) The histogram shows relative mRNA expression levels at 3D and was normalized to those of α-ACTIN mRNA levels, then to similarly normalized values for 0D tissues. Primer sequences used are listed in Materials and Methods (**b**) Line graphs represent expression profile changes between L-clod- and L-PBS-treated groups. (**c**) Changes in protein expression at different time points were assessed by ELISA and were represented in dot plots (line at mean). All data are presented as mean ± SD (*n* ≥ 3. * *p* < 0.05, ** *p* < 0.01, *** *p* < 0.001.).

**Figure 5 ijms-23-11287-f005:**
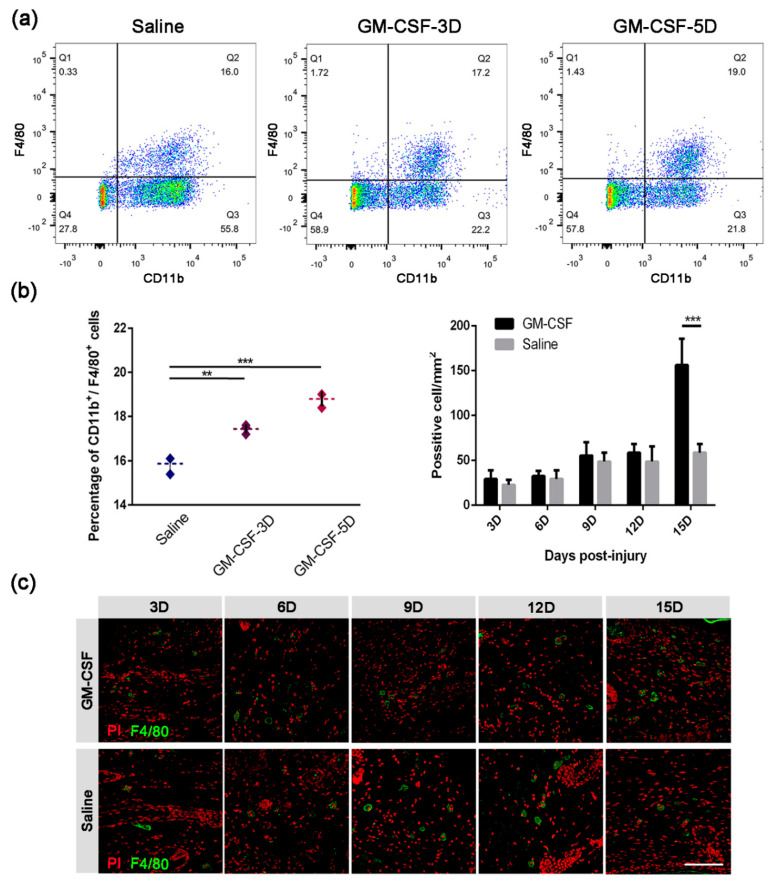
**Quantification of macrophages in peripheral blood and wound tissue after treatment with GM-CSF.** (**a**) Flow cytometry analysis of macrophages in peripheral blood based on surface marker CD11b and F4/80. The statistics of the two positive cells are presented as mean ± SD in subfigure (**b**), **left** (*n* ≥ 3. ** *p* < 0.01, *** *p* < 0.001). (**c**) IF staining of tissue sample sections from either GM-CSF- or saline-injected mice, using anti-F4/80 antibody (scale bar: 100 µm). The statistics of F4/80 positive cells in the high-power field are presented as mean ± SD in subfigure (**b**), **right** (*n* ≥ 3. ** *p* < 0.01, *** *p* < 0.001).

**Figure 6 ijms-23-11287-f006:**
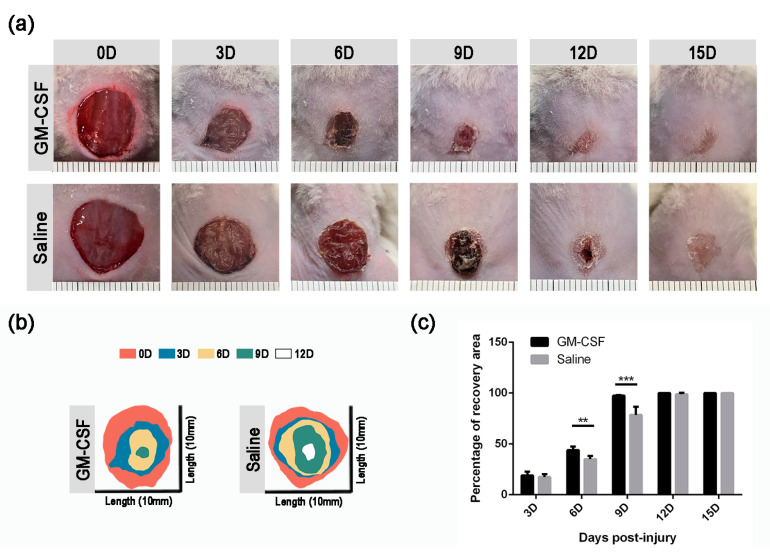
**Wound healing of saline- and GM-CSF-injected mice.** (**a**) Macroscopic appearance of wound sites from GM-CSF-treated and control mice. (**b**) Schematic diagram of injury site changes over time. (**c**) Quantification of the percentage of recovery area rate (*n* ≥ 3. ** *p* < 0.01, *** *p* < 0.001. Data presented as mean ± SD).

**Figure 7 ijms-23-11287-f007:**
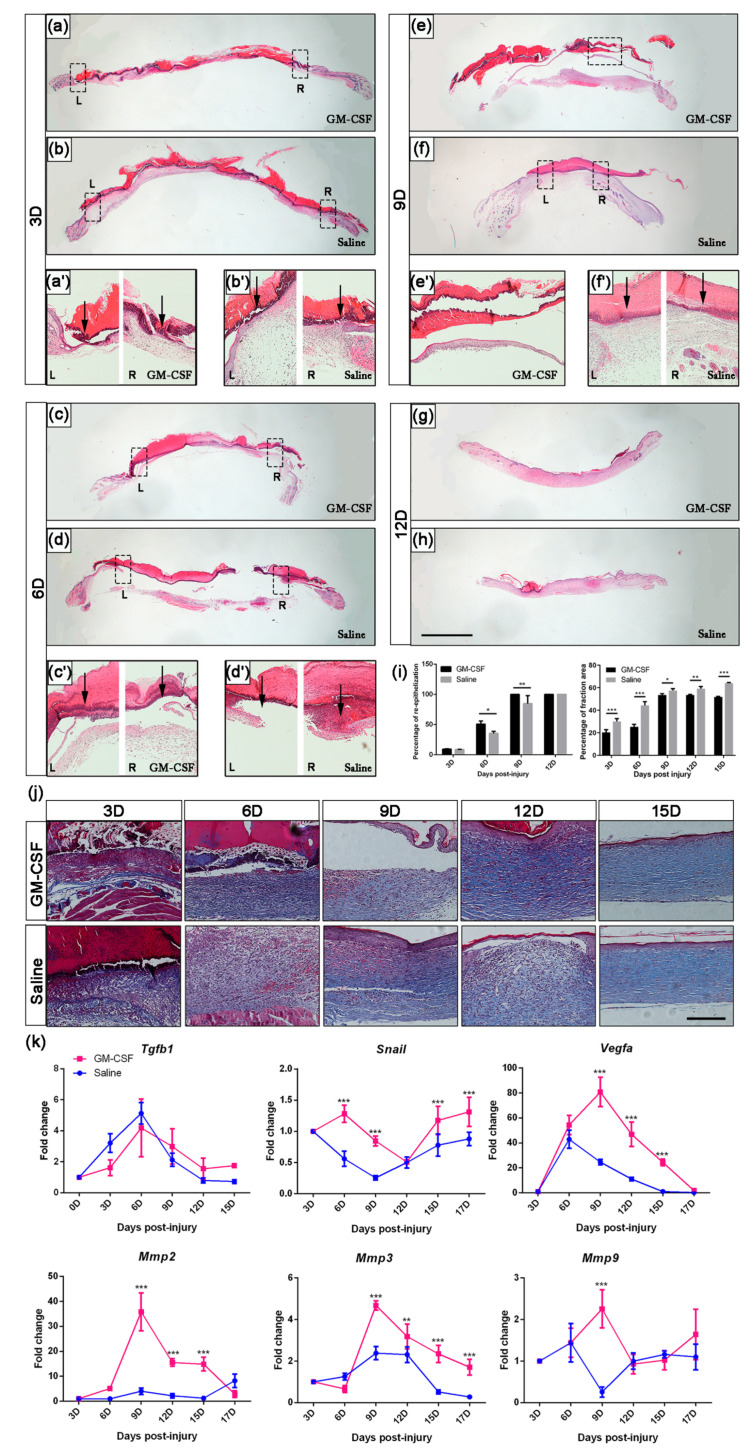
**Analysis of the effect of GM-CSF treatment during wound healing process.** (**a**–**h**) Re-epithelialization analysis of GM-CSF- vs. saline-injected mice at 3D, 6D, 9D, 12D, and 15D (scale bar: 2.5 mm). Dashed boxes mark inset showing high-magnification images (**a’**–**f’**). Black arrows indicate ends of migrating epithelial tongues. The statistics of quantification of the percentage of re-epithelialization are presented as mean ± SD in subfigure (**i**), **left** (*n* ≥ 3. * *p* < 0.05, ** *p* < 0.01). (**j**) Collagen deposition analysis and the statistics are shown in subfigure (**i**), **right** (scale bar: 200 µm. *n* ≥ 3. * *p* < 0.05, ** *p* < 0.01, *** *p* < 0.001. Data presented as mean ± SD). (**k**) Gene expression profiles at different time points. (*n* ≥ 3. ** *p* < 0.01, *** *p* < 0.001. Data presented as mean ± SD).

**Table 1 ijms-23-11287-t001:** Statistics of healing rate in L-PBS- and L-clod-injected mice.

Time Points	Groups		*p* Value
	L-Clod-Injected	L-PBS-Injected	
5D	20.43 ± 2.49%	33.71 ± 1.89%	<0.001
10D	66.31 ± 4.04%	95.24 ± 1.66%	<0.001
15D	97.34 ± 2.13%	100.00 ± 0.00%	0.13
20D	100.00 ± 0.00%	100.00 ± 0.00%	>0.05

Data presented as mean ± SD.

**Table 2 ijms-23-11287-t002:** Statistics of healing rate in Saline- and GM-CSF-injected mice.

Time Points	Groups		*p* Value
	Saline-Injected	GM-CSF-Injected	
3D	17.44 ± 2.92%	19.05 ± 3.67%	0.56
6D	35.00 ± 1.93%	43.98 ± 2.01%	<0.001
9D	78.40 ± 4.64%	97.38 ± 0.42%	<0.001
12D	98.83 ± 1.53%	100.00 ± 0.00%	0.67
15D	100.00 ± 0.00%	100.00 ± 0.00%	>0.05

Data presented as mean ± SD.

**Table 3 ijms-23-11287-t003:** Primer sequences.

Gene Product		
CSF1R	**Forward**	5′-GAAGGTGGCTGTGAAGATGCTAA-3′
	**Reverse**	5′-AGGTTGACTATATTCTCGTGCTGTC-3′
VEGFA	**Forward**	5′-AGGCTGCTGTAACGATGAAG-3′
	**Reverse**	5′-TCTCCTATGTGCTGGCTTTG-3′
TGFβ1	**Forward**	5′-CTGAACCAAGGAGACGGAATAC-3′
	**Reverse**	5′-GGGCTGATCCCGTTGATTT-3′
IL6	**Forward**	5′-CTTCCATCCAGTTGCCTTCT-3′
	**Reverse**	5′-CTCCGACTTGTGAAGTGGTATAG-3′
SNAIL	**Forward**	5′-GAGAAGCCATTCTCCTGCTC-3′
	**Reverse**	5′-GCACTGGTATCTCTTCACATCC-3′
CAMP	**Forward**	5′-TCCCTAGACACCAATCTCTACC-3′
	**Reverse**	5′-GCCACATACAGTCTCCTTCAC-3′
MMP3	**Forward**	5′-ATGTCACTGGTACCAACCTATTC-3′
	**Reverse**	5′-CAAGTCTGTGGAGGACTTGTAG-3′
MMP9	**Forward**	5′-TGCACTGGGCTTAGATCATTC-3′
	**Reverse**	5′-TGCCGTCTATGTCGTCTTTATTC-3′
MMP2	**Forward**	5′-CTGGAATGCCATCCCTGATAA-3′
	**Reverse**	5′-GGTTCTCCAGCTTCAGGTAATAA-3′
FGF2	**Forward**	5′-CTTACCGGTCACGGAAATACTC-3′
	**Reverse**	5′-AGCTCTTAGCAGACATTGGAAG-3′
TNFα	**Forward**	5′-CGATGGGTTGTACCTTGTCTAC-3′
	**Reverse**	5′-GCAGAGAGGAGGTTGACTTTC-3′
IL8	**Forward**	5′-TCCTGCTTGAATGGCTTGAATACTA-3′
	**Reverse**	5′-CGGTGTCCTGATTATCGTCCTC-3′
α-ACTIN	**Forward**	5′-CTCCCTGGAGAAGAGCTATGA-3′
	**Reverse**	5′-CCAAGAAGGAAGGCTGGAAA-3′

## Data Availability

The datasets used and/or analyzed during the current study are available from the corresponding author on reasonable request.

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
