# Peer review of "The Stimulation of Macrophages by Systematical Administration of GM-CSF Can Accelerate Adult Wound Healing Process"

_ijms, 2022, doi:10.3390/ijms231911287_

Round 1
Reviewer 1 Report
The article presents some interesting results, but some changes are needed to improve the understanding of the work. My suggestions are:
Line 47: the authors must add some references.
Figure 1b-c: The authors should change the letters in the caption because are wrong and change the description of the lower d graph.
Figure 1d lower: To show that the L-clod really affect the inhibition of macrophages is better to introduce in the lower graph also the column related to the control skin lesion, which will be like the L-PBS injection.
Line 107: From the image used in figure 1c also L-PBS seems to decrease the presence of macrophages with respect to the control lesion.
How can the authors comment on that result?
Line 124: Is better to rewrite this sentence, because the re-epithelization process is not the closure of the wound, indeed the authors, in the results reported upper, indicated that the closure of the wound is 15D after injection of L-PBS and 20 days after L-clod injection.
Figure 5a: The authors should indicate the marker tested in the axis of FACS analysis to better understand the analysis performed and how to interpret the results
Figure 5b-c: From the description of the results is not so clear how many injections of the GM-CSF were performed. Are they two and the evaluation was 3 days post the second injection, or what? In the second part of the figure, was the analysis of the presence of the macrophages performed after only one injection?
Why also did the authors change the evaluation time point with respect to the starting experiments?
Line 264: Is better to introduce a reference here.
Line 360: The injection of GM-CSF will be better described, mostly in terms of how much time the injection was performed, because from the results reported before it is understood that the injection was performed a maximum of two times.
Also another question for the authors: Can the authors hypothesize the long-term advantages and disadvantages of repeated injections of GM-CSF? And also why the authors did not evaluate the effect of a single injection of GM-CSF?
Reviewer 2 Report
The authors studied "The stimulation of macrophages by systematical administration of GM-CSF can accelerate adult wound healing process.
Healing of skin wounds, especillay the diificult-to-heal-ones, is an interesting topic
I do not see the novelty of this paper. It is a well-known fact that macrophages participate in wound healing. Experiments in mice were done previously. It is also known that macrophages are necessary in all phages of healing, especially during inflammation and the effect of granulocyte-macrophage colony-stimulating factor is also known. Results with macrophages-depleted mice were published by some authors previously.
Histology should be described by marks to show what is there.
What about M2 macrophages?
The problem is the impact of macrophages in chronic wounds.
Round 2
Reviewer 2 Report
I recommend accepting the paper in current form.